# Understanding Batch Normalization

**Johan Bjorck, Carla Gomes, Bart Selman, Kilian Q. Weinberger**
Cornell University
{njb225,gomes,selman,kqw4} @cornell.edu

## Abstract

Batch normalization (BN) is a technique to normalize activations in intermediate layers of deep neural networks. Its tendency to improve accuracy and speed up training have established BN as a favorite technique in deep learning. Yet, despite its enormous success, there remains little consensus on the exact reason and mechanism behind these improvements. In this paper we take a step towards a better understanding of BN, following an empirical approach. We conduct several experiments, and show that BN primarily enables training with larger learning rates, which is the cause for faster convergence and better generalization. For networks without BN we demonstrate how large gradient updates can result in diverging loss and activations growing uncontrollably with network depth, which limits possible learning rates. BN avoids this problem by constantly correcting activations to be zero-mean and of unit standard deviation, which enables larger gradient steps, yields faster convergence and may help bypass sharp local minima. We further show various ways in which gradients and activations of deep unnormalized networks are ill-behaved. We contrast our results against recent findings in random matrix theory, shedding new light on classical initialization schemes and their consequences.

## 1 Introduction

Normalizing the input data of neural networks to zero-mean and constant standard deviation has been known for decades [29] to be beneficial to neural network training. With the rise of deep networks, Batch Normalization (BN) naturally extends this idea across the intermediate layers within a deep network [23], although for speed reasons the normalization is performed across mini-batches and not the entire training set. Nowadays, there is little disagreement in the machine learning community that BN accelerates training, enables higher learning rates, and improves generalization accuracy [23] and BN has successfully proliferated throughout all areas of deep learning [2, 17, 21, 46]. However, despite its undeniable success, there is still little consensus on why the benefits of BN are so pronounced. In their original publication [23] Ioffe and Szegedy hypothesize that BN may alleviate "internal covariate shift" – the tendency of the distribution of activations to drift during training, thus affecting the inputs to subsequent layers. However, other explanations such as improved stability of concurrent updates [13] or conditioning [42] have also been proposed.

Inspired by recent empirical insights into deep learning [25, 36, 57], in this paper we aim to clarify these vague intuitions by placing them on solid experimental footing. We show that the activations and gradients in deep neural networks without BN tend to be heavy-tailed. In particular, during an early on-set of divergence, a small subset of activations (typically in deep layer) "explode". The typical practice to avoid such divergence is to set the learning rate to be sufficiently small such that no steep gradient direction can lead to divergence. However, small learning rates yield little progress along flat directions of the optimization landscape and may be more prone to convergence to sharp local minima with possibly worse generalization performance [25].

BN avoids activation explosion by repeatedly correcting all activations to be zero-mean and of unit standard deviation. With this "safety precaution", it is possible to train networks with large learning

rates, as activations cannot grow incrontrollably since their means and variances are normalized. SGD with large learning rates yields faster convergence along the flat directions of the optimization landscape and is less likely to get stuck in sharp minima.

We investigate the interval of viable learning rates for networks with and without BN and conclude that BN is much more forgiving to very large learning rates. Experimentally, we demonstrate that the activations in deep networks without BN grow dramatically with depth if the learning rate is too large. Finally, we investigate the impact of random weight initialization on the gradients in the network and make connections with recent results from random matrix theory that suggest that traditional initialization schemes may not be well suited for networks with many layers — unless BN is used to increase the network's robustness against ill-conditioned weights.

## 1.1 The Batch Normalization Algorithm

As in [23], we primarily consider BN for convolutional neural networks. Both the input and output of a BN layer are four dimensional tensors, which we refer to as $I_{b,c,x,y}$ and $O_{b,c,x,y}$, respectively. The dimensions corresponding to examples within a batch $b$, channel $c$, and two spatial dimensions $x, y$ respectively. For input images the channels correspond to the RGB channels. BN applies the same normalization for all activations in a given channel,

$$O_{b,c,x,y} \leftarrow \gamma_c \frac{I_{b,c,x,y} - \mu_c}{\sqrt{\sigma_c^2 + \epsilon}} + \beta_c \qquad \forall\, b, c, x, y. \tag{1}$$

Here, BN subtracts the mean activation $\mu_c = \frac{1}{|\mathcal{B}|} \sum_{b,x,y} I_{b,c,x,y}$ from all input activations in channel $c$, where $\mathcal{B}$ contains all activations in channel $c$ across all features $b$ in the entire mini-batch and all spatial $x, y$ locations. Subsequently, BN divides the centered activation by the standard deviation $\sigma_c$ (plus $\epsilon$ for numerical stability) which is calculated analogously. During testing, running averages of the mean and variances are used. Normalization is followed by a channel-wise affine transformation parametrized through $\gamma_c, \beta_c$, which are learned during training.

## 1.2 Experimental Setup

To investigate batch normalization we will use an experimental setup similar to the original Resnet paper [17]: image classification on CIFAR10 [27] with a 110 layer Resnet. We use SGD with momentum and weight decay, employ standard data augmentation and image preprocessing techniques and decrease learning rate when learning plateaus, all as in [17] and with the same parameter values. The original network can be trained with initial learning rate 0.1 over 165 epochs, however which fails without BN. We always report the best results among initial learning rates from $\{0.1, 0.003, 0.001, 0.0003, 0.0001, 0.00003\}$ and use enough epochs such that learning plateaus. For further details, we refer to Appendix B in the online version [4].

## 2 Disentangling the benefits of BN

Without batch normalization, we have found that the initial learning rate of the Resnet model needs to be decreased to $\alpha = 0.0001$ for convergence and training takes roughly 2400 epochs. We refer to this architecture as an unnormalized network. As illustrated in Figure 1 this configuration does not attain the accuracy of its normalized counterpart. Thus, seemingly, batch normalization yields faster training, higher accuracy and enable higher learning rates. To disentangle how these benefits are related, we train a batch normalized network using the learning rate and the number of epochs of an unnormalized network, as well as an initial learning rate of $\alpha = 0.003$ which requires 1320 epochs for training. These results are also illustrated in Figure 1, where we see that a batch normalized networks with such a low learning schedule performs no better than an unnormalized network. Additionally, the train-test gap is much larger than for normalized networks using lr $\alpha = 0.1$, indicating more overfitting. A learning rate of $\alpha = 0.003$ gives results in between these extremes. This suggests that it is the higher learning rate that BN enables, which mediates the majority of its benefits; it improves regularization, accuracy and gives faster convergence. Similar results can be shown for variants of BN, see Table 4 in Appendix K of the online version [4].

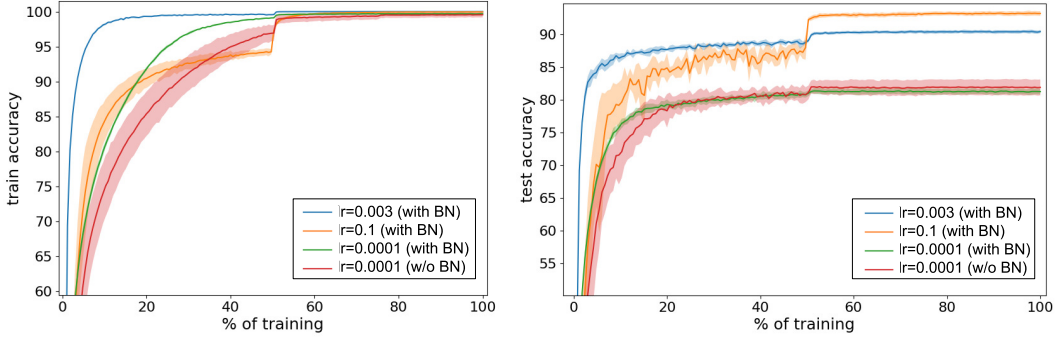

**Figure 1:** The training (*left*) and testing (*right*) accuracies as a function of progress through the training cycle. We used a 110-layer Resnet with three distinct learning rates 0.0001, 0.003, 0.1. The smallest, 0.0001 was picked such that the network without BN converges. The figure shows that with matching learning rates, both networks, with BN and without, result in comparable testing accuracies (red and green lines in right plot). In contrast, larger learning rates yield higher test accuracy for BN networks, and diverge for unnormalized networks (not shown). All results are averaged over five runs with std shown as shaded region around mean.

## 2.1 Learning rate and generalization

To explain these observations we consider a simple model of SGD; the loss function $\ell(x)$ is a sum over the losses of individual examples in our dataset $\ell(x) = \frac{1}{N}\sum_{i=1}^{N}\ell_i(x)$. We model SGD as sampling a set $B$ of examples from the dataset with replacements, and then with learning rate $\alpha$ estimate the gradient step as $\alpha\nabla_{SGD}(x) = \frac{\alpha}{|B|}\sum_{i\in B}\nabla\ell_i(x)$. If we subtract and add $\alpha\nabla\ell(x)$ from this expression we can restate the estimated gradient $\nabla_{SGD}(x)$ as the true gradient, and a noise term

$$\alpha\nabla_{SGD}(x) = \underbrace{\alpha\nabla\ell(x)}_{\text{gradient}} + \underbrace{\frac{\alpha}{|B|}\sum_{i\in B}\big(\nabla\ell_i(x) - \nabla\ell(x)\big)}_{\text{error term}}.$$

We note that since we sample uniformly we have $\mathbb{E}\big[\frac{\alpha}{|B|}\sum_{i\in B}\big(\nabla\ell_i(x) - \nabla\ell(x)\big)\big] = 0$. Thus the gradient estimate is unbiased, but will typically be noisy. Let us define an architecture dependent noise quantity $C$ of a single gradient estimate such that $C = \mathbb{E}\big[\|\nabla\ell_i(x) - \nabla\ell(x)\|^2\big]$. Using basic linear algebra and probability theory, see Apppendix D, we can upper-bound the noise of the gradient step estimate given by SGD as

$$\mathbb{E}\big[\|\alpha\nabla\ell(x) - \alpha\nabla_{SGD}(x)\|^2\big] \leq \frac{\alpha^2}{|B|}C. \tag{2}$$

Depending on the tightness of this bound, it suggests that the noise in an SGD step is affected similarly by the learning rate as by the inverse mini-batch size $\frac{1}{|B|}$. This has indeed been observed in practice in the context of parallelizing neural networks [14, 49] and derived in other theoretical models [24]. It is widely believed that the noise in SGD has an important role in regularizing neural networks [6, 57]. Most pertinent to us is the work of Keskar et al. [25], where it is empirically demonstrated that large mini-batches lead to convergence in sharp minima, which often generalize poorly. The intuition is that larger SGD noise from smaller mini-batches prevents the network from getting "trapped" in sharp minima and therefore bias it towards wider minima with better generalization. Our observation from (2) implies that SGD noise is similarly affected by the learning rate as by the inverse mini-bath size, suggesting that a higher learning rate would similarly bias the network towards wider minima. We thus argue that the better generalization accuracy of networks with BN, as shown in Figure 1, can be explained by the higher learning rates that BN enables.

## 3 Batch Normalization and Divergence

So far we have provided empirical evidence that the benefits of batch normalization are primarily caused by higher learning rates. We now investigate why BN facilitates training with higher learning rates in the first place. In our experiments, the maximum learning rates for unnormalized networks

have been limited by the tendency of neural networks to *diverge* for large rates, which typically happens in the first few mini-batches. We therefore focus on the gradients at initialization. When comparing the gradients between batch normalized and unnormalized networks one consistently finds that the gradients of comparable parameters are larger and distributed with heavier tails in unnormalized networks. Representative distributions for gradients within a convolutional kernel are illustrated in Figure 2.

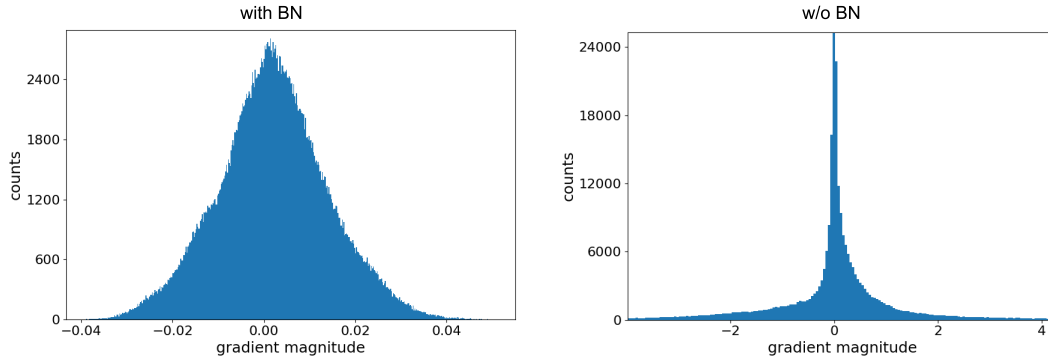

**Figure 2:** Histograms over the gradients at initialization for (midpoint) layer 55 of a network with BN (*left*) and without (*right*). For the unnormalized network, the gradients are distributed with heavy tails, whereas for the normalized networks the gradients are concentrated around the mean. (Note that we have to use different scales for the two plots because the gradients for the unnormalized network are almost two orders of magnitude larger than for the normalized on.)

A natural way of investigating divergence is to look at the loss landscape along the gradient direction during the first few mini-batches that occur with the normal learning rate (0.1 with BN, 0.0001 without). In Figure 3 we compare networks with and without BN in this regard. For each network we compute the gradient on individual batches and plot the relative change in loss as a function of the step-size (i.e. new_loss/old_loss). (Please note the different scales along the vertical axes.) For unnormalized networks only small gradient steps lead to reductions in loss, whereas networks with BN can use a far broader range of learning rates.

Let us define *network divergence* as the point when the loss of a mini-batch increases beyond $10^3$ (a point from which networks have never managed to recover to acceptable accuracies in

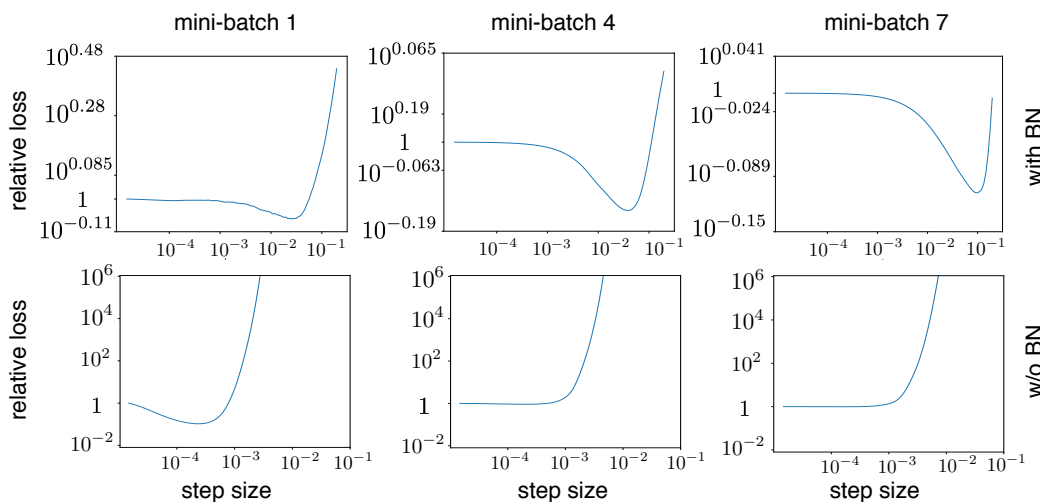

**Figure 3:** Illustrations of the relative loss over a mini-batch as a function of the step-size (normalized by the loss before the gradient step). Several representative batches and networks are shown, each one picked at the start of the standard training procedure. Throughout all cases the network with BN (bottom row) is far more forgiving and the loss decreases over larger ranges of $\alpha$. Networks without BN show divergence for larger step sizes.

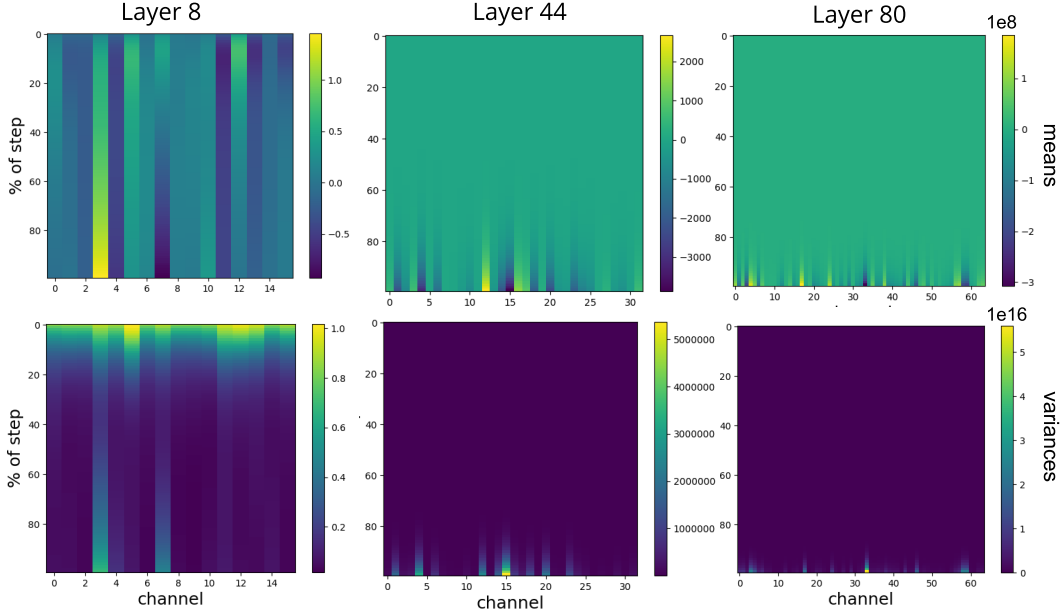

**Figure 4:** Heatmap of channel means and variances during a diverging gradient update (without BN). The vertical axis denote what percentage of the gradient update has been applied, $100\%$ corresponds to the endpoint of the update. The moments explode in the higher layer (note the scale of the color bars).

our experiments). With this definition, we can precisely find the gradient update responsible for divergence. It is interesting to see what happens with the means and variances of the network activations along a 'diverging update'. Figure 4 shows the means and variances of channels in three layers (8,44,80) during such an update (without BN). The color bar reveals that the scale of the later layer's activations and variances is orders of magnitudes higher than the earlier layer. This seems to suggest that the divergence is caused by activations growing progressively larger with network depth, with the network output "exploding" which results in a diverging loss. BN successfully mitigates this phenomenon by correcting the activations of each channel and each layer to zero-mean and unit standard deviation, which ensures that large activations in lower levels cannot propagate uncontrollably upwards. We argue that this is the primary mechanism by which batch normalization enables higher learning rates. This explanation is also consistent with the general folklore observations that shallower networks allow for larger learning rates, which we verify in Appendix H. In shallower networks there aren't as many layers in which the activation explosion can propagate.

## 4 Batch Normalization and Gradients

Figure 4 shows that the moments of unnormalized networks explode during network divergence and Figure 5 depicts the moments as a function of the layer depth after initialization (without BN) in log-scale. The means and variances of channels in the network tend to increase with the depth of the network even at initialization time — suggesting that a substantial part of this growth is data independent. In Figure 5 we also note that the network transforms normalized inputs into an output that reaches scales of up to $10^2$ for the largest output channels. It is natural to suspect that such a dramatic relationship between output and input are responsible for the large gradients seen in Figure 2. To test this intuition, we train a Resnet that uses one batch normalization layer only at the very last layer of the network, normalizing the output of the last residual block but no intermediate activation. Such an architecture allows for learning rates up to $0.03$ and yields a final test accuracy of $90.1\%$, see Appendix E — capturing two-thirds of the overall BN improvement (see Figure 1). This suggests that normalizing the final layer of a deep network may be one of the most important contributions of BN.

For the final output layer corresponding to the classification, a large channel mean implies that the network is biased towards the corresponding class. In Figure 6 we created a heatmap of $\frac{\partial L_b}{\partial O_{b,j}}$ after

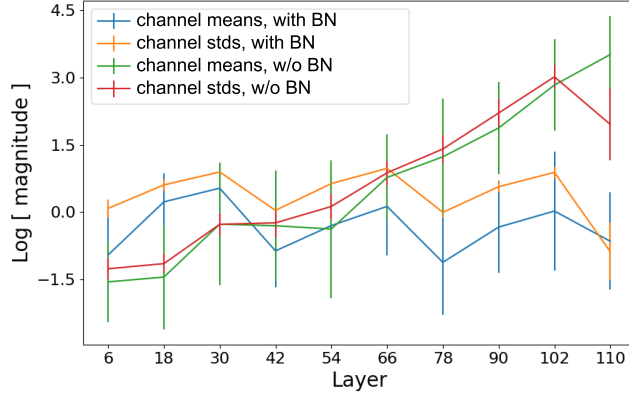

**Figure 5:** Average channel means and variances as a function of network depth at initialization (error bars show standard deviations) on log-scale for networks with and without BN. The batch normalized network the mean and variances stays relatively constant throughout the network. For an unnormalized network, they seem to grow almost exponentially with depth.

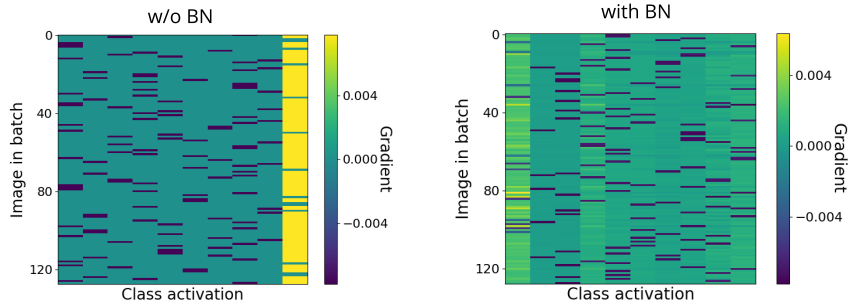

**Figure 6:** A heat map of the output gradients in the final classification layer after initialization. The columns correspond to a classes and the rows to images in the mini-batch. For an unnormalized network (*left*), it is evident that the network consistently predicts one specific class (very right column), irrespective of the input. As a result, the gradients are highly correlated. For a batch normalized network, the dependence upon the input is much larger.

initialization, where $L_b$ is the loss for image $b$ in our mini-batch, and activations $j$ corresponds to class $j$ at the final layer. A yellow entry indicates that the gradient is positive, and the step along the negative gradient would decrease the prediction strength of this class for this particular image. A dark blue entry indicates a negative gradient, indicating that this particular class prediction should be strengthened. Each row contains one dark blue entry, which corresponds to the true class of this particular image (as initially all predictions are arbitrary). A striking observation is the distinctly yellow column in the left heatmap (network without BN). This indicates that after initialization the network tends to almost always predict the same (typically wrong) class, which is then corrected with a strong gradient update. In contrast, the network with BN does not exhibit the same behavior, instead positive gradients are distributed throughout all classes. Figure 6 also sheds light onto why the gradients of networks without BN tend to be so large in the final layers: the rows of the heatmap (corresponding to different images in the mini-batch) are highly correlated. Especially the gradients in the last column are positive for almost all images (the only exceptions being those image that truly belong to this particular class label). The gradients, summed across all images in the minibatch, therefore consist of a sum of terms with matching signs and yield large absolute values. Further, these gradients differ little across inputs, suggesting that most of the optimization work is done to rectify a bad initial state rather than learning from the data.

## 4.1 Gradients of convolutional parameters

We observe that the gradients in the last layer can be dominated by some arbitrary bias towards a particular class. Can a similar reason explain why the gradients for convolutional weights are larger

| | $a = \sum_{bxy} \lvert d^{bxy}_{c_o c_i ij} \rvert$ | $b = \lvert \sum_{bij} d^{bxy}_{c_o c_i ij} \rvert$ | $a/b$ |
|---|---|---|---|
| Layer 18, with BN | 7.5e-05 | 3.0e-07 | **251.8** |
| Layer 54, with BN | 1.9e-05 | 1.7e-07 | **112.8** |
| Layer 90, with BN | 6.6e-06 | 1.6e-07 | **40.7** |
| Layer 18, w/o BN | 6.3e-05 | 3.6e-05 | **1.7** |
| Layer 54, w/o BN | 2.2e-04 | 8.4e-05 | **2.6** |
| Layer 90, w/o BN | 2.6e-04 | 1.2e-04 | **2.1** |

**Table 1:** Gradients of a convolutional kernel as described in (4) at initialization. The table compares the absolute value of the sum of gradients, and the sum of absolute values. Without BN these two terms are similar in magnitude, suggesting that the summands have matching signs throughout and are largely data independent. For a batch normalized network, those two differ by about two orders of magnitude.

for unnormalized networks. Let us consider a convolutional weight $K_{o,i,x,y}$, where the first two dimensions correspond to the outgoing/ingoing channels and the two latter to the spatial dimensions. For notational clarity we consider 3-by-3 convolutions and define $S = \{-1, 0, 1\} \times \{-1, 0, 1\}$ which indexes into $K$ along spatial dimensions. Using definitions from section 1.1 we have

$$O_{b,c,x,y} = \sum_{c'} \sum_{x',y' \in S} I_{b,c',x+x',y+y'} K_{c,c',x',y'} \tag{3}$$

Now for some parameter $K_{o,i,x,y}$ inside the convolutional weight $K$, its derivate with respect to the loss is given by the backprop equation [40] and (3) as

$$\frac{\partial L}{\partial K_{o,i,x',y'}} = \sum_{b,x,y} d^{bxy}_{o,i,x',y'}, \qquad \text{where} \qquad d^{bxy}_{o,i,x',y'} = \frac{\partial L}{\partial O_{b,o,x,y}} I_{b,i,x+x',y+y'}. \tag{4}$$

The gradient for $K_{o,i,x,y}$ is the sum over the gradients of examples within the mini-batch, and over the convoluted spatial dimensions. We investigate the signs of the summands in (4) across both network types and probe the sums at initialization in Table 1. For an unnormalized networks the absolute value of (4) and the sum of the absolute values of the summands generally agree to within a factor 2 or less. For a batch normalized network, these expressions differ by a factor of $10^2$, which explains the stark difference in gradient magnitude between normalized and unnormalized networks observed in Figure 2. These results suggest that for an unnormalized network, the summands in (4) are similar across both spatial dimensions and examples within a batch. They thus encode information that is neither input-dependent or dependent upon spatial dimensions, and we argue that the learning rate would be limited by the large input-independent gradient component and that it might be too small for the input-dependent component. We probe these questions further in Appendix J, where we investigate individual parameters instead of averages.

Table 1 suggests that for an unnormalized network the gradients are similar across spatial dimensions and images within a batch. It's unclear however how they vary across the input/output channels $i, o$. To study this we consider the matrix $\mathbf{M}_{i,o} = \sum_{xy} \lvert \sum_{bxy} d^{bxy}_{oixy} \rvert$ at initialization, which intuitively measures the average gradient magnitude of kernel parameters between input channel $i$ and output channel $o$. Representative results are illustrated in Figure 7. The heatmap shows a clear trend that some channels constantly are associated with larger gradients while others have extremely small gradients by comparison. Since some channels have large means, we expect in light of (4) that weights outgoing from such channels would have large gradients which would explain the structure in Figure 7. This is indeed the case, see Appendix G in the online version [4].

## 5   Random initialization

In this section argue that the gradient explosion in networks without BN is a natural consequence of random initialization. This idea seems to be at odds with the trusted Xavier initialization scheme [12] which we use. Doesn't such initialization guarantee a network where information flows smoothly between layers? These initialization schemes are generally derived from the desiderata that the variance of channels should be constant when randomization is taken over random weights. We argue that this condition is too weak. For example, a pathological initialization that sets weights to 0 or

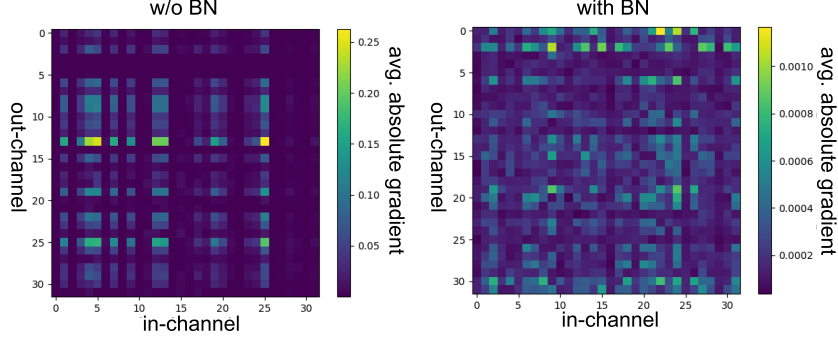

**Figure 7:** Average absolute gradients for parameters between in and out channels for layer 45 at initialization. For an unnormalized network, we observe a dominant low-rank structure. Some in/out-channels have consistently large gradients while others have consistently small gradients. This structure is less pronounced with batch normalization (*right*).

100 with some probability could fulfill it. In [12] the authors make simplifying assumptions that essentially result in a linear neural network. We consider a similar scenario and connect them with recent results in random matrix theory to gain further insights into network generalization. Let us consider a simple toy model: a linear feed-forward neural network where $A_t \dots A_2 A_1 x = y$, for weight matrices $A_1, A_2 \dots A_n$. While such a model clearly abstracts away many important points they have proven to be valuable models for theoretical studies [12, 15, 32, 56]. CNNs can, of course, be flattened into fully-connected layers with shared weights. Now, if the matrices are initialized randomly, the network can simply be described by a product of random matrices. Such products have recently garnered attention in the field of random matrix theory, from which we have the following recent result due to [30].

**Theorem 1** *Singular value distribution of products of independent Gaussian matrices [30]. Assume that $X = X_1 X_2 \dots X_M$, where $X_i$ are independent $N \times N$ Gaussian matrices s.t. $\mathbb{E}[X_{i,jk}] = 0$ and $\mathbb{E}[X_{i,jk}^2] = \sigma_i^2/N$ for all matrices $i$ and indices $j, k$. In the limit $N \to \infty$, the expected singular value density $\rho_M(x)$ of $X$ for $x \in \left(0, (M+1)^{M+1}/M^M\right)$ is given by*

$$\rho_M(x) = \frac{1}{\pi x}\frac{\sin((M+1)\varphi)}{\sin(M\varphi)}\sin\varphi, \qquad where \qquad x = \frac{\left(\sin((M+1)\varphi)\right)^{M+1}}{\sin\varphi(\sin(M\varphi))^M} \qquad (5)$$

Figure 8 illustrates some density plots for various values of $M$ and $\theta$. A closer look at (5) reveals that the distribution blows up as $x^{-M/(M+1)}$ nears the origin, and that the largest singular value scales as $\mathcal{O}(M)$ for large matrices. In Figure 9 we investigate the singular value distribution for practically sized matrices. By multiplying more matrices, which represents a deeper linear network, the singular values distribution becomes significantly more heavy-tailed. Intuitively this means that the ratio between the largest and smallest singular value (the condition number) will increase with depth, which we verify in Figure 20 in Appendix K.

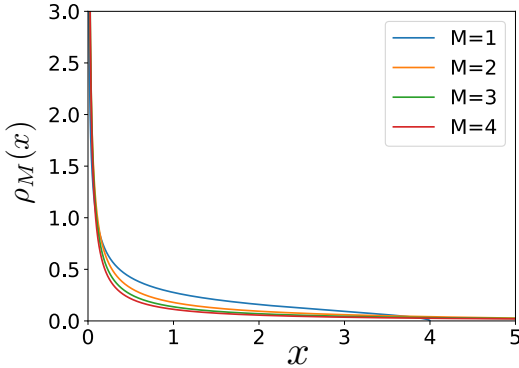

**Figure 8:** Distribution of singular values according to theorem 1 for some $M$. The theoretical distribution becomes increasingly heavy-tailed for more matrices, as does the empirical distributions of Figure 9

Consider $\min_{A_i,\ i=1,2\ \dots\ t} \|A_t \dots A_2 A_1 x - y\|^2$, this problem is similar to solving a linear system $\min_x \|Ax - y\|^2$ if one only optimizes over a single weight matrix $A_i$. It is well known that the complexity of solving $\min_x \|Ax - y\|$ via gradient descent can be characterized by the condition number $\kappa$ of $A$, the ratio between largest $\sigma_{max}$ and smallest singular value $\sigma_{min}$. Increasing

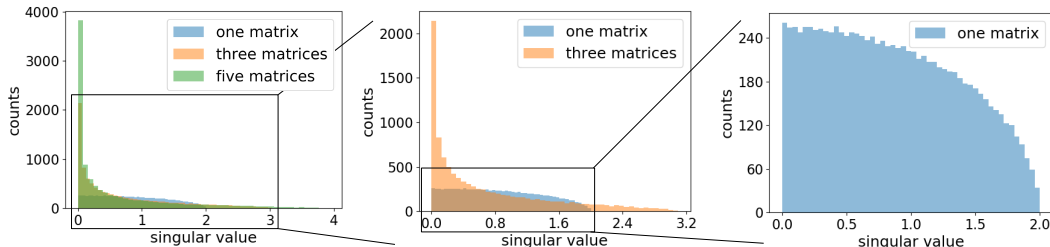

**Figure 9:** An illustration of the distributions of singular values of random square matrices and product of independent matrices. The matrices have dimension N=1000 and all entries independently drawn from a standard Gaussian distribution. Experiments are repeated ten times and we show the total number of singular values among all runs in every bin, distributions for individual experiments look similar. The left plot shows all three settings. We see that the distribution of singular values becomes more heavy-tailed as more matrices are multiplied together.

$\kappa$ has the following effects on solving a linear system with gradient descent: **1)** convergence becomes slower, **2)** a smaller learning rate is needed, **3)** the ratio between gradients in different subspaces increases [3]. There are many parallels between these results from numerical optimization, and what is observed in practice in deep learning. Based upon Theorem 1, we expect the conditioning of a linear neural network at initialization for more shallow networks to be better which would allow a higher learning rate. And indeed, for an unnormalized Resnet one can use a much larger learning if it has only few layers, see Appendix H. An increased condition number also results in different subspaces of the linear regression problem being scaled differently, although the notion of subspaces are lacking in ANNs, Figure 5 and 7 show that the scale of channels differ dramatically in unnormalized networks. The Xavier [12] and Kaming initialization schemes [16] amounts to a random matrix with iid entries that are all scaled proportionally to $n^{-1/2}$, the same exponent as in Theorem 1, with different constant factors. Theorem 1 suggests that such an initialization will yield ill-conditioned matrices, independent of these scale factors. If we accept these shortcomings of Xavier-initialization, the importance of making networks robust to initialization schemes becomes more natural.

## 6   Related Work

The original batch normalization paper posits that internal covariate explains the benefits of BN [23]. We do not claim that internal covariate shift does not exist, but we believe that the success of BN can be explained without it. We argue that a good reason to doubt that the primary benefit of BN is eliminating internal covariate shift comes from results in [34], where an initialization scheme that ensures that all layers are normalized is proposed. In this setting, internal covariate shift would not disappear. However, the authors show that such initialization can be used instead of BN with a relatively small performance loss. Another line of work of relevance is [48] and [47], where the relationship between various network parameters, accuracy and convergence speed is investigated, the former article argues for the importance of batch normalization to facilitate a phenomenon dubbed 'super convergence'. Due to space limitations, we defer discussion regarding variants of batch normalization, random matrix theory, generalization as well as further related work to Appendix A in the online version [4].

## 7   Conclusions

We have investigated batch normalization and its benefits, showing how the latter are mainly mediated by larger learning rates. We argue that the larger learning rate increases the implicit regularization of SGD, which improves generalization. Our experiments show that large parameter updates to unnormalized networks can result in activations whose magnitudes grow dramatically with depth, which limits large learning rates. Additionally, we have demonstrated that unnormalized networks have large and ill-behaved outputs, and that this results in gradients that are input independent. Via recent results in random matrix theory, we have argued that the ill-conditioned activations are natural consequences of the random initialization.

**Acknowledgements**

We would like to thank Yexiang Xue, Guillaume Perez, Rich Bernstein, Zdzislaw Burda, Liam McAllister, Yang Yuan, Vilja Järvi, Marlene Berke and Damek Davis for help and inspiration. This research is supported by NSF Expedition CCF-1522054 and Awards FA9550-18-1-0136 and FA9550-17-1-0292 from AFOSR. KQW was supported in part by the III-1618134, III-1526012, IIS-1149882, IIS-1724282, and TRIPODS- 1740822 grants from the National Science Foundation, and generous support from the Bill and Melinda Gates Foundation, the Office of Naval Research, and SAP America Inc.

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
