[Reviews · NeurIPS 2018]

Reviewer 1



The main argument is that: 1. Sec 2: convergence and generalization in NNs are separate issues. Convergence is necessary for generalizing well, but if a network converges without normalization, BN does not add further improvement in generalization. 2. Sec. 3 & 4: convergence is governed by random initialization and gradient magnitudes (e.g. exploding gradient problem, the importance of condition number not growing exponentially w.r.t. depth). Interestingly, the distribution of gradient magnitudes between layer units in an unnormalized network has high variance (i.e. poor conditioning) but the variance of gradients *between* minibatch inputs is small, suggesting that the gradient is not depending on the inputs at all and has been overwhelmed by biases introduced by poor initialization. 3. Sec 3 & 4 experiments: BN fixes convergence problems for higher learning rate settings, by keeping the condition number roughly unitary. In contrast, unnormalized networks cannot train with large learning rates because gradients would scale exponentially. 4. Under the "low noise == sharp minima == poor generalization" hypothesis, using larger learning rates scales up the "SGD noise" (Eq 2), which leads to better generalization. Strengths: - The authors approach an important area of study in theoretical understanding of Deep Learning: why does batch normalization improve training times, convergence on training set, and generalization to test set? - Reproducibility of experiments via code included in supplemental material. - Empirical variance analysis of gradients w.r.t. intra-layer and intra-batch. - Showing that the unnormalized network has a low-rank gradients was intruiiging. - Empirical rigour Weaknesses: - My general reservation about this paper is that while it was helpful in clarifying my own understanding of BN, a lot of the conclusions are consistent with folk wisdom understanding of BN (e.g. well-conditioned optimization), and the experimental results were not particularly surprising. Questions: - Taking Sharp Minima Hypothesis at face value, Eq 2 suggests that increasing gradient variance improves generalization. This is consistent with the theme that decreasing LR or decreasing minibatch size make generalization worse. Can you comment on how to reconcile this claim with the body of work in black-box optimization (REBAR, RELAX, VIMCO, Reinforcement Learning) suggesting that *reducing* variance of gradient estimation improves generalization & final performance? - Let's suppose that higher SGD variance (eq 2) == better generalization. BN decreases intra-unit gradient variance (Fig 2, left) but increases intra-minibatch variance (Fig 4, right). When it is applied to a network that converges for some pair \alpha and B, it seems to generalize slightly worse (Fig 1, right). According to the explanations presented by this paper, this would imply that BN decreased M slightly. For what unnormalized architectures does the application of BN increase SGD variance, and for what unnormalized architectures does BN actually decrease SGD variance? (requiring LR to be increased to compensate?) How do inter-layer gradient variance and inter-minibatch gradient variance impact on generalization? - For an unnormalized network, is it possible to converge AND generalize well by simply using a small learning rate with a small batch size? Does this perform comparably to using batch norm? - While Section 4 was well-written, the argument that BN decreases exponential condition number is not new; this situation has been analyzed in the case of training deep networks using orthogonal weights (https://arxiv.org/pdf/1511.06464.pdf, https://arxiv.org/pdf/1806.05393.pdf), and the exploding / vanishing gradient problem (Hochreiter et al.). On the subject of novelty, does this paper make a stronger claim than existing folk wisdom that BN makes optimization well-conditioned?

Reviewer 2



In this submission, the authors undertake an empirical study of batch normalization, in service of providing a more solid foundation for why the technique works. The authors study a resnet trained on CIFAR-10, with and without batch norm (BN) to draw their conclusions. They first point out that BN enables training with faster learning rates, and then argue that this allows SGD to have a greater regularizing effect (via reference to the flat minima results). The authors then put forward a sequence of measurements that argue that without normalization, the optimization problem is poorly conditioned. This includes showing that: the gradient distribution is heavy tailed (Fig 2); the output at initialization is biased to a single class (and so are the gradients; Fig 4); the gradients in the convolutions spatially align at initialization, producing a strong input-independent term (Table 1); the gradients in the convolutions align across channels at initialization, producing low-rank structure (Fig 5). In each case, BN ameliorates the effect. These effects combine with a final appeal to random matrix theory that the condition number of the unnormalized network (ignoring nonlinearities) is almost certainly very poor in expectation, and that BN does something to help. I like this paper: it's topical (given the Rahimi appeal), generally well-written (though there's some room for improvement), suggestive, and well explored. I don't think it's airtight, and there are a few errors scattered about that I think are fixable. Even if it doesn't turn out to be the full explanation for BN success, it's a step in the right direction, and likely to have a significant impact on the field -- both in terms of its content, and its general approach. Some corrections or clarifications required: - L60: "BN has a lower training loss indicating overfitting". It has neither a lower train or test loss from what I can tell in Fig 1, and not sure why lower train => overfitting. - Fig 1: legends are hard to parse, put in actual learning rates in better order - some mistakes in Section 2.1 equations. L73: there should be an summation in the expectation. Eqn (2): should be a square in the denominator (||B||^2). The expression in L74 of the "noise of the gradient estimate" would be better expressed as the noise in the gradient step, since the gradient estimate itself won't have the factor of \alpha in it. - L115: I think this should refer to Fig 4, not Fig 3. But then Fig 4 is showing the gradients, not the activations. I found Fig 4 confusing as the x-axis says "class activation", but this is really "class". Given that the text talks about classes being favoured, it might be worthwhile to separately show the class activations (showing that one class is favoured), and the gradients (showing that these are imbalanced). - One other point on Fig 4: it seems like an obvious solution is just to immediately adjust the class biases at initialization to equalize average class assignments. I think the authors aren't doing this because they're making the claim in the next section that this is a general phenomenon throughout the network (where it can't be fixed by adjusting biases). It would be worth pointing out that this is an illustration. - Table 1 legend, last line "We also see that...". I cannot see this trend within the data in the table. - L196: equation (7)? - L216: "Result Theorem 1"? - L218: "changing entry distribution"?

Reviewer 3



The paper tries to demystify the success of BatchNorm, which despite widespread use does not have a good explanation why or how it works. The manuscript separates into two somewhat unrelated parts: 1. The authors claim that most or all of the benefits of BN are mediated via enabling the use of a larger learning rate. This is backed up experimentally by showing BN with a small LR does not perform appreciably better than no Normalization. 2. The authors try empirically to identify the mechanism by which BN enables the larger learning rates. The authors mostly succeed on arguing the first point, but fail to cite the work of Leslie Smith, who has done rather in depth work on why, when it comes to learning rate, larger is generally better, and how various forms of regularization such as L2 interact with it. Similarly, Smith and Le recently analyzed learning rate in a Bayesian framework. Note that this analysis contradicts the earlier statement from Keskar about "large batch sizes lead to sharp minima" which is cited here, so please include a more balanced discussion of previous work. Further, this observation does not help to explain why existing alternatives to BN, such as LayerNorm, GroupNorm, InstanceNorm etc. don't work as well. Investigating these alternatives is an active field given limitations of BN to deal with small batches, and being computationally expensive. Would methods like Batch Renormalization or GroupNorm be expected to work equally well if they enable and equally large LR? In my experience they work without changes to learning rate, so why don’t they? In the second part, the authors go beyond the standard approach of analyzing convergence via Eigenvalues of the Jacobian (which don't scale beyond small toy examples), by showing that there is a data independent component dominating the gradient in the "no normalization" case. This is an intriguing finding that I think they authors did not study in enough depth. First of all, it would be useful to study where this gradient direction is pointing us. E.g. aggregating and averaging over a few batches should give a clean estimate of this term, and it could be tested if it always points in the radial direction rather than along the hypersphere of constant radius. If we follow this direction, does it lead to a reduction in cost, or is it a very narrow minimum with regions of high cost lurking closely behind it? Since this is only a single dimension in a high dimensional space, it could be insightful to provide visualizations along this direction, compared to e.g. a random orthogonal direction. This leads to the testable hypothesis, that gradients in the radial direction are large and constrain the learning rate, while gradients tangential to the hypersphere are much smaller. Minor comments: - 72, typo, "let define" should be "let us define" - line 75, equation 2: Please provide a bit more detail what linear algebra goes into deriving this equation. See the similar derivation in Smith and Le, "A Bayesian Perspective on Generalization and Stochastic Gradient Descent", section 5, which arrives at a linear scaling rule. - 79, typo, "lead" should be "lead to" - line 105, worded a bit ambiguously, please make clear it's a single BN layer, not on top of each residual block. - Table 1 is hard to parse, might be better as a bar plot - 175: Are you aware of the related work by Schönholz and Sohl-Dickstein, e.g. https://arxiv.org/abs/1611.01232 and https://arxiv.org/abs/1806.05393 about initalization in deep linear networks? The latter introduces the tf.ConvolutionDeltaOrthogonal, which is similar to the ideas here. - 186 typo: "largest singular" missing "value"